# Experimental Infection of Domestic Pigs (*Sus scrofa*) with Rift Valley Fever Virus

**DOI:** 10.3390/v15020545

**Published:** 2023-02-16

**Authors:** Baratang Alison Lubisi, Paidamwoyo Barry Mutowembwa, Phumudzo Nomicia Ndouvhada, Lieza Odendaal, Armanda D. S. Bastos, Mary-Louise Penrith

**Affiliations:** 1Agricultural Research Council—Onderstepoort Veterinary Research, Onderstepoort, Pretoria 0110, South Africa; 2Department of Zoology and Entomology, Faculty of Natural and Agricultural Sciences, University of Pretoria, Hatfield, Pretoria 0028, South Africa; 3Department of Paraclinical Sciences, Faculty of Veterinary Science, University of Pretoria, Onderstepoort, Pretoria 0110, South Africa; 4Department of Veterinary Tropical Diseases, Faculty of Veterinary Science, University of Pretoria, Onderstepoort, Pretoria 0110, South Africa

**Keywords:** pathology, polymerase chain reaction, serology, sequencing, phylogenetics

## Abstract

Rift valley fever (RVF), caused by the RVF virus (RVFV), is a vector-borne zoonotic disease that primarily affects domestic ruminants. Abortion storms and neonatal deaths characterise the disease in animals. Humans develop flu-like symptoms, which can progress to severe disease. The susceptibility of domestic pigs (*Sus scrofa domesticus*) to RVFV remains unresolved due to conflicting experimental infection results. To address this, we infected two groups of pregnant sows, neonates and weaners, each with a different RVFV isolate, and a third group of weaners with a mixture of the two viruses. Serum, blood and oral, nasal and rectal swabs were collected periodically, and two neonates and a weaner from group 1 and 2 euthanised from 2 days post infection (DPI), with necropsy and histopathology specimens collected. Sera and organ pools, blood and oronasorectal swabs were tested for RVFV antibodies and RNA. Results confirmed that pigs can be experimentally infected with RVFV, although subclinically, and that pregnant sows can abort following infection. Presence of viral RNA in oronasorectal swab pools on 28 DPI suggest that pigs may shed RVFV for at least one month. It is concluded that precautions should be applied when handling pig body fluids and carcasses during RVF outbreaks.

## 1. Introduction

Rift valley fever (RVF), first described by [1], is a vector-borne zoonotic disease, which primarily affects domestic ruminants and camelids. It is caused by RVF virus (RVFV), in the Bunyavirales order, *Phenuiviridae* family and *Phlebovirus* genus [2]. Animals are predominantly infected through the bite of infected mosquitoes, but vertical transmission is possible [3]. Transmission to humans can occur through contact with aerosolised virus during handling and opening of infected carcasses, and to a lesser extent via mosquito bites. The disease is characterised by abortion storms and neonatal deaths in animals, while humans normally present with self-limiting flu-like signs. However, the disease can progress to severe hepatic disease with haemorrhagic manifestations, renal impairment, encephalitis, ocular complications and death [4,5,6]. Diagnosis of RVF employs antibody and antigen detection methods including virus isolation, virus neutralisation, RT-PCR, ELISA or histopathology with immunohistochemistry [7]. Outbreaks of RVF may have serious economic impacts due to imposed trade bans and devastating health consequences for both humans and livestock. Vaccination of livestock and decrease of mosquito populations through integrated vector management practices are used to prevent and control disease outbreaks [8,9].

Effective outbreak response and mitigation activities rely on vaccination, risk mapping, predictive models, using satellites and climate data, and early warning systems (EWS) in RVF-prone areas [10]. However, effective EWS development with increased predictive accuracy requires both climatic and non-climatic information [10]. Non-climatic factors include indicators of the vulnerability of populations to disease outbreaks and include host exposure to viruses, susceptibility to infection, immune response and virus adaptation, shedding and spread. The two major obstacles for determining natural host ranges of viruses are incomplete field investigations and the inability to segregate hosts essential for prolonged biological transmission from others. This is because the degrees of contribution to viral transmission are not the same among competent hosts [11].

Pigs are integral to Africa’s mixed species farming systems. They come into contact with humans, are bitten by mosquitoes, scavenge and eat dead animals including aborted foetuses, and have been shown to sero-convert following natural infection with RVFV based on studies conducted in Egypt [1,12,13,14]. However, the role of domestic pigs (*Sus scrofa*) in the epidemiology of RVF has not been thoroughly investigated. Research conducted in the 1950s and 1960s provided conflicting and circumstantial evidence of porcine susceptibility to RVFV [15], leading to the general assumption that the species was refractory to RVFV infection. Scientists, however, concede that there was minimal information available for conclusive assertions to be made on domestic pig susceptibility and their potential role in RVFV maintenance [16,17]. Knowledge gaps in the epidemiology of RVF with special reference to pig susceptibility to RVFV led to a recent study in which weaners were successfully infected with the ZH501 strain of RVFV, as proven by seroconversion, virus isolation from sera and oronasal swabs and RNA isolation from the isolates [16]. These latest findings warrant additional efforts to establish the effects of RVFV on pig neonates and pregnant sows, in the interest of determining their potential role in virus maintenance and transmission, and if necessary, devising improved prevention and control strategies. This is particularly important in light of increasing urbanisation and swine populations in Sub-Saharan Africa [18].

Until the recent study by [16], early experiments to determine swine susceptibility to RVFV only involved inoculating fewer than five pigs of undisclosed ages and mostly measuring their temperatures and determining viraemia or antibody presence, without providing details of the tests used [1,15,19,20]. The aim of the present study was to address shortcomings of previous studies through experimental infection of pregnant sows at different gestation periods, neonatal piglets of less than one week and weaners. Ante and post mortem samples from the pigs were periodically collected, and analysed using serological and antigen detection assays for a period of sixty days. A further aim was to inform decisions on whether to include or exclude this species in non-climatic EWS information in pig farming areas of Africa that experience atypical RVF outbreaks, thus clarifying their relevance in the epidemiology of the disease.

## 2. Materials and Methods

### 2.1. Virus Strains and Cell Culture

Two RVF viruses isolated at the Agricultural Research Council—Onderstepoort Veterinary Research (ARC—OVR) were used in the challenge experiments. Virus 1 (a variant of M66/09) was a bovine liver isolate from the 2009 RVF outbreak in the Gauteng Province of South Africa, used at passage 5BHK.15Vero and a titre of 5 × 10^7^ pfu/mL. Virus 2 (a variant of M21/10) was an ovine organ pool isolate from the 2010 outbreak in the Free State Province of South Africa, used at passage 3BHK.14Vero and a titre of 1.5 × 10^5^ pfu/mL. The viruses were propagated in Vero cells (ATCC, Manassas, VA, USA) maintained with Dulbecco’s Modified Eagle’s Medium (DMEM) (Life Technologies, Carlsbad, CA, USA) containing 2% foetal bovine serum (FBS) (Sigma-Aldrich, Saint Louis, MO, USA) and 1× Penicillin and Streptomycin each (Gibco, New Yolk, NY, USA) and incubated at 37 °C with 5% CO_2_. Partial nucleotide sequencing of the glycoprotein gene (Gn) encoded by the M-genome segment was performed for the two viruses prior to serial passaging in organ culture for purposes of verifying their genotypes. Strains M66/09 and M21/10 belong to two different S, M and L segment genotypes [21]. The use of two genetically diverse isolates was opted for in case one was more amenable to establishing infection in the pig host than the other.

### 2.2. Animals and Experimental Design

Animal experiments were performed following appropriate acclimatisation periods at a biological safety level 3 (BSL3) animal facility at ARC—OVR Transboundary Animal Diseases Programme (TADP), under animal ethics committee (AEC) approval numbers AEC10.16 and EC057-17. Large white pregnant sows (PS: n = 9), lactating sows (LS: n = 3), 1–3-day-old suckling piglets (SP: n = 30) and 6–8-week-old weaners (W: n = 27) were obtained from the ARC Animal Production Institute in Irene, Gauteng Province. Black head Dorper lambs (L: n = 8), 1–2 weeks old, and their dams, the ewes (E: n = 8), were sourced from a commercial farm in the Northern Cape Province. The animals were divided into three groups (1–3), and inoculated with virus 1, virus 2 and a mixture of both virus 1 and virus 2, respectively. Group 1 animals (n = 29) were housed in stables A and B, group 2 (n = 29) in stables C and D and group 3 (n = 17) in stable E. Accommodation, animal identities and treatment regimens are detailed in Appendix A.

Two suckling piglets each in stables A and C, one weaner each in stables B and D, and two weaners each in stable E served as mock inoculation controls and were infected with 2 mL of tissue culture medium intravenously (i/v). No lactating animal was inoculated. All remaining animals in group 1 (stables A and B) and group 2 (stables C and D) were inoculated with 2 mL of virus 1 (i/v) and virus 2 (i/v), respectively. In group 3 (stable E), all four lambs and three weaners were inoculated (i/v) with 2 mL of a mixture of virus 1 and virus 2 (1:1 *v*/*v*), and two weaners were inoculated with 2 mL (i/v) of virus 1 and virus 2 each. Mixing of the two isolates was performed to investigate the difference, if any, co-infection would make to achieving successful infection and resultant clinical course in the pig model, compared to a single virus.

The animals were monitored for discomfort and clinical signs twice daily, and temperatures were recorded every day. Normal pig and sheep temperatures were regarded as 38.7–39.8 °C and 38.3–39.9 °C, respectively [22]. Resting temperatures of 38 °C were also recorded as normal since the animals did not show signs of discomfort. Scoring of clinical signs was performed per species (Table 1 and Table 2). Sera, blood and oral, nasal and rectal swabs were collected at days post infection (DPI) 0 to 7, then at 14, 21, 28 and 60 DPI if the animal was still alive.

Two suckling piglets and one weaner were randomly selected, starting with the infected ones, and euthanised in groups 1 and 2 by intracardiac injection with a barbiturate overdose (Eutha-naze, Bayer Health Care, Animal health, Johannesburg, South Africa) every two days from DPI 2, while the remaining pigs were euthanised on DPI 60. Lactating sows were euthanised when there were no suckling piglets left, by stunning with a captive bolt pistol followed by severing of the carotid artery to ensure death. Pregnant sows were euthanised following termination of pregnancy or farrowing using the same method utilised for lactating sows. Their newborn piglets were given an overdose of a barbiturate intracardially after birth. The weaners in group 3 were euthanised on DPI 30. Euthanasia of the lambs in all groups was indicated when they were too ill to feed and interact with their surroundings normally, as per experimental end-point scores approved by the animal ethics committee, or on DPI 30, using the same method applied in suckling piglets and weaners.

### 2.3. Laboratory Tests

#### 2.3.1. Serology

The competitive ELISA kit for the detection of anti-Rift Valley fever (RVF) antibodies in ruminant serum or plasma (ID Screen^®^ Rift valley fever Competition Multi-species, Louis Pasteur, Paris, France) was used for RVFV antibody detection. The assay is a multispecies test applicable for use on ruminants, horses, dogs and other species. Porcine and ovine sera from all experiments (n = 495) were tested according to the manufacturer’s instructions. Sera with Sample/Negative percentage (S/N%) less than or equal to 40 were regarded as positive, those between 40 and 50 were deemed doubtful and samples above 50 were considered negative for RVFV antibodies. For the purpose of this study, all doubtful results were regarded as positive.

#### 2.3.2. Virus Isolation

Virus isolation was performed on 1/10 suspensions of pooled organs (n = 85) and terminal blood (n = 64) samples of all pregnant sows and their offspring, and pooled organs (n = 47) and terminal bleeds (n = 33) of control lambs and ewes, lactating sows, suckling piglets and weaners (n = 47), from groups 1, 2 and 3, using standard methods [7].

#### 2.3.3. Viral RNA Extraction, Real-Time and Conventional RT-PCR

Total RNA from blood, oral, nasal and rectal swab pools, and homogenated organ samples, was extracted at the ARC-OVR Biotechnology PCR Laboratory using the magnetic-bead capture MagMAX-96 total RNA Isolation kit (MagNA Pure LC Instrument, Roche, Johannesburg, South Africa). A published real-time reverse transcriptase-polymerase chain reaction (RT-PCR) assay was used to test blood (n = 140), pooled organs (n = 107) and oronasorectal swabs (n = 83) from infected pregnant sows and their offspring. Blood (n = 168), pooled organs (n = 59) and oronasorectal swabs (n = 193) of pigs and control lambs from experiments involving infection of suckling piglets and weaners, and uninfected ewes and lactating sows, were also tested [23].

For conventional RT-PCR, nucleic acid extraction was performed using E.Z.N.A Viral RNA kit (Omegabio-tek, Norcross, GA, USA) and TRIzol reagent (Invitrogen & Thermofisher, Johannesburg, South Africa) according to the respective manufacturer’s instructions. The RT-PCR method of [24] was used to test tissue-cultured (TC) organ pool material from the three infection groups, constituting pregnant sows (n = 9), newborn piglets (n = 76), weaners (n = 9), ewes (n = 2) and lambs (n = 4), using a OneStep RT-PCR kit (QIAGEN, Germantown, MD, USA). The amplicons were mixed with 2 µL loading dye (Promega, Madison, WI, USA), loaded on a 1% agarose gel (Lonza, Fair Lawn, NJ, USA) containing 2 µL ethidium bromide (Promega, Madison, WI, USA), together with a molecular weight marker of 1.5 kb (Promega, Madison, WI, USA), electrophoresed at 120 volts for 20–30 min and then visualised under UV light for identification of positive samples (551 bp amplicons).

#### 2.3.4. Viral RNA Sequencing and Phylogenetic Analysis

The correct size amplicons generated from the conventional RT-PCR (n = 17) were purified directly from the tube using the Roche High Pure PCR Product Purification Kit (Roche Diagnostics, Johannesburg, South Africa). Bidirectional Sanger sequencing was performed on clean products with each of the PCR primers in separate reactions using the BigDye Terminator Cycle Sequencing Ready Reaction kit (Applied Biosystem, Johannesburg, South Africa) and submitted to the core Sanger sequencing facility of the University of Pretoria (Gauteng, South Africa).

Sequence chromatograms were edited and uploaded in the basic local alignment search tool (BLAST) for identification and selection with closely related nucleotide sequences available in the Genbank database [25]. Sequences were aligned using ClustalW in MEGAX [26] and end-unaligned regions were trimmed prior to generating summary statistics in MEGAX. The final dataset (353 nucleotides in length) was used to infer a neighbour-joining tree [27], using the best-fit model identified under the Bayesian Information Criterion, with 10,000 bootstrap replicates performed to evaluate nodal support [28].

#### 2.3.5. Pathology

Post mortems were conducted in the post mortem hall of the BSL3 animal facility following death of all experimental animals. Organ, blood and serum samples were collected for demonstration of anti-RVFV antibodies, RVFV RNA detection, virus isolation, histopathological examination, anti-RVFV immunohistochemistry and electron microscopic imaging.

For histopathology, liver, spleen and kidney samples collected in 10% neutral buffered formalin were embedded in paraffin wax using the standard protocol of the histopathology laboratory at the University of Pretoria, Faculty of Veterinary Science. Histopathology lesions were scored according to species.

Immunohistochemistry for RVFV antigens was conducted on duplicate tissue sections using a polyclonal mouse ascitic fluid (National Institute for Communicable Disease,
Johannesburg, Sandringham, South Africa) and an avidin-biotinylated peroxidase complex (ABC) immunodetection technique, as previously described [29]. Briefly, the standard immunoperoxidase method included routine deparaffination with two changes of xylene, rehydration through graded alcohol baths to distilled water and incubation with 3% hydrogen peroxide for 15 min. This was followed by heat-induced epitope retrieval in citrate buffer (pH 6.0), followed by incubation with the anti-RVFV primary antibody (1:500) for 1 h. Sections were sequentially incubated with the rabbit-anti-mouse secondary antibody (F0232, DakoCytomation, Glostrup, Denmark), followed by detection with a standard avidin-biotin peroxidase system, Vectastain^®^ Elite^®^ ABC-HRP Kit (PK-6100, Vector Laboratories, Inc., Newark, CA, USA), NovaRED peroxidase substrate (SK-4800, Vector Laboratories, Inc., Newark, CA, USA) and haematoxylin counterstain. Slides were examined for positive labelling, typified as fine diffuse to coarse granular cytoplasmic brownish labelling using a light microscope. All microscopic images were captured with a DP25 camera (Olympus, Tokyo, Japan) on a light microscope (BX46 Olympus, Tokyo, Japan) using standard software (CellSens Version 1.12 Olympus, Tokyo, Japan).

#### 2.3.6. Electron Microscopy

The livers of aborted foetuses 1, 2, 5 and 10 from pregnant sow 5 in stable A were homogenised in PBS (1/10) and centrifuged at 3000 rpm for 15 min, their supernatants collected and centrifuged at 13,000 rpm for 45 min, with the resulting supernatant discarded and a drop of double distilled water poured on the sediment, followed by a drop of phosphotungstic acid. Standard negative staining transmission electron microscopy (TEM) techniques for identification of RVF virions were performed at the electron microscopy unit at the University of Pretoria with a few modifications [30].

### 2.4. Statistical Analysis

Differences in values of key experimental parameters such as proportion seropositive, temperatures, clinical and histopathological scores and Ct-values between groups were compared statistically. Paired data were analysed using paired *t*-test and independent datasets were evaluated using unpaired *t*-test and Mann–Whitney U Test [31,32]. Differences in proportions and binary datasets were evaluated using a comparison of proportions calculator and Fisher’s exact test, respectively [33].

## 3. Results

### 3.1. Clinical Signs

#### 3.1.1. Pregnant Sows and Offspring

There was one abortion 10 days before the expected farrowing date in a group 1 sow infected with the M66/09 virus variant (Figure 1). The remaining sows in groups 1 and those in group 2 infected with M21/10 virus variant, farrowed 1 to 7 days before the expected date and did not display overt clinical signs or discomfort. Rectal temperatures in both groups remained within the normal range of 38–39.8 °C (*p* > 0.05) (Figure 2). Temperatures of newborn piglets (P) were not recorded in both groups, but stillborns, neonatal deaths, small and weak piglets and those with congenital abnormalities were observed (Figure 1). Median clinical scores for groups 1 and 2 were 0.52 and 1.7 respectively, and their distribution did not differ significantly (Mann–Whitney U = 6, group 1 ≠ group 2, *p* = 0.4, two-tailed), and neither did they differ significantly from those of their respective control lambs *(p >* 0.05).

#### 3.1.2. Suckling Piglets and Weaners

No overt clinical signs were observed among infected suckling piglets and weaners in groups 1 and 2, and those infected with a mixture of M66/09 and M21/10 virus variants in group 3. Their temperatures remained within the normal range, except for slight pyrexia observed in a few animals, mainly on 1 DPI, including one control suckling piglet in group 1 (Table 2; Figure 2). Mean temperature differences within and between groups were not significant *(p >* 0.05). Suckling piglets and weaners in groups 1 and 2 had median clinical scores of 1 and 0 respectively, which did not vary significantly (Mann–Whitney U = 160, group 1 = group 2, *p* = 0.17, two-tailed). Group 2 pigs and their respective control lambs had median clinical scores of 0 and 3 respectively, whose distribution varied significantly (Mann–Whitney U = 32, group 1 ≠ group 2, *p* = 0.016, two-tailed).

#### 3.1.3. Control Lambs, Ewes and Lactating Sows

Infection control lambs in groups 1 (n = 2) and group 3 (n = 4) showed pyrexia for the first 5 DPI and on 6 DPI, respectively. Group 2 lambs showed severe clinical signs without temperature rises and were euthanised on 3 DPI (Figure 2; Table 2). Uninfected lactating sow and ewe (n = 1 each) in group 1 had fluctuating temperature rises above 40 °C between 2 and 7 DPI, while the ewes in group 3 (n = 4) showed the same results between 1 and 11 DPI.

### 3.2. Pathology

#### 3.2.1. Macroscopic Observations

##### Pregnant Sows and Resultant Offspring

Pregnant sows did not show gross macroscopic lesions or abnormalities, but a few external observations were made from piglets that were born ill and weak, and those that died shortly after birth, either from natural causes or euthanasia. The affected animals had smaller carcasses compared to litter mates (group 1: n = 2; group 2: n = 5) and poor condition scores of approximately 1.5/5 to 2/5 (group 2: n = 6). Arthrogryposis (group 2: n = 2), splay legs with associated decubitus ulcers (group 2: n = 3) and umbilical hernia (group 1: n = 1) were observed (Figure 2). Internal lesions of varying severity and distribution patterns were seen and mostly involved the liver, kidney and spleen (Figure 3). Congestion and a few haemorrhages were the main observations associated with the gastrointestinal tract (GIT). Almost all major organ systems of the aborted foetuses (group 1) and stillborns (group 2) exhibited lesions (Figure 3).

The lesions observed in the foetuses and stillborn pigs are described below.

Central nervous system: cerebral and cerebellar hypoplasia (group 1: n = 1); brains with jelly-like and semi-liquefied cerebrum and cerebellum (group 1: n = 10), and those with no clear delineation between the grey and white matter (group 2: n = 2); congestion of the brain, meninges and blood vessels (group 1: n = 5; and group 2: n = 4); and a pink, soft and friable spinal cord (group 1: n = 1) (Figure 3). Other than cerebral and cerebellar hypoplasia, these observations may be attributable to post mortem changes.

Circulatory system: pallor of the myocardium (group 1: n = 5; group 2: n = 3), haemorrhages (group 1: n = 8) and congestion (group 1: n = 1); blood-tinged (group 1: n = 4; and group 2: n = 1) and straw-coloured (group 1: n = 3; group 2: n = 1) hydrothorax were present also in a neonate (group 1: n = 1) (Figure 3); clear (group 1: n = 2) and blood-tinged (group 2: n = 1) ascitic fluid in the abdominal cavities and haemoperitoneum (group 1: n = 1) (Figure 3). Many of these observations may be attributable to post mortem changes, but those involving haemorrhage or transudates in body cavities may also be due to RVFV infection.

Respiratory system: lung lesions included oedema (group 1: n = 2; group 2: n = 1), congestion and pulmonary consolidation (hepatisation) coupled with oedema in some (group 1: n = 5; group 2: n = 1) and a combination of oedema, congestion and haemorrhage (group 1: n = 4; group 2: n = 1) (Figure 3). The changes are highly likely to be RVFV infection related.

Digestive and hepatobiliary systems: liver lesions varied in degrees of severity and different distribution patterns. Regular post mortem changes included friability, dark red discolouration and congestion. These findings are common in ovine foetuses [34] and may be post mortem changes in porcines as well. Lesions that may be attributable to RVFV infection include haemorrhages, hepatic necrosis (1–2 mm) (group 1: n = 4) and pallor or diffuse yellow discoloration (group 2: n = 2) (Figure 3). In the GIT, congestion of the mucosa was the only post mortem finding (group 1: n = 8; group 2, n = 3) (Figure 3).

Urogenital system: Kidney lesions in the aborted foetuses (group 1) included enlargement, friability, congestion, haemorrhages and infarcts (1–5 mm). Capsules were hard to peel over the necrotic areas, leaving rough surfaces. The kidneys of stillborn pigs were pale and pulpy and one was severely congested with an infarct (group 2: n = 4). The testicles (group 1: n = 2) were very small and muscles (group 1: n = 2) showed generalised congestion. The congestion and friability may be post mortem changes.

Immune system: Spleens were pale-pink and friable (group 1: n = 8) and haemorrhagic (group 1: n = 1); others displayed haemorrhages and redness, pallor and pulpiness, as well as haemorrhages and infarcts (group 2, n = 3). Similar to the urogenital system, the colour changes and friability may be post mortem changes.

##### Suckling Piglets and Weaners

Lesions were mostly seen on the liver, spleen and kidneys. These organs presented with congestion, haemorrhage and necrosis of varying severity and distribution (Figure 3).

#### 3.2.2. Histopathology and Immunohistochemistry

Histopathological lesions were assigned scores as described in Table 3, and the scores were compared among the different treatment groups (Table 4). Histopathological examinations were performed on liver (n = 153), kidney (n = 150) and spleen (n = 150) samples, and only a limited number of livers (n = 76), spleens (n = 21) and kidneys (n = 11) were subjected to IHC testing. The IHC signals were faint, most probably due to the low RVF antigen concentrations in the analysed tissues. Since uninfected in-contact pigs analysed were proven to be horizontally infected in this study, there were no negative pig tissue controls for comparison, and the observed faint IHC signals could be false positives.

##### Pregnant Sows and Resultant Offspring

The liver of one sow per group was analysed and only the sow in group 2 showed hepatocyte swelling. Aborted piglets or those born from infected sows in groups 1 and 2 displayed lesions as described in Table 4 and Figure 4. Both groups had a median histopathological score of 1 and the distributions did not differ significantly (Mann–Whitney U = 948, group 1 ≠ group 2, *p* = 0.269, two-tailed). However, significant median distribution differences described by Mann–Whitney U = 71.5, group 1 ≠ group 2, *p* = 0.02, two-tailed, and Mann–Whitney U = 2, group 1 ≠ group 2, *p* = 0.0, two-tailed, were observed between group 1 porcines (median = 1) and control lambs (median = 2) and group 2 porcines (median = 1) and control lambs (median = 0), respectively. A few livers (n = 6) tested positive on IHC (Table 4; Figure 4). Development of hepatocyte glycogen storage vacuoles in hepatocytes is also normal in fed pigs [36].

##### Suckling Piglets and Weaners

Lesions observed in suckling piglets and weaners in all three groups are described in Table 4. The median histopathological scores for groups 1 and 2 were both 1 and their distributions were not significantly different (Mann–Whitney U = 152, group 1 ≠ group 2, *p* = 0.103, two-tailed). Significant differences in the median score distributions of group 1 pigs (median = 1) and their control lambs (median = 2) (Mann–Whitney U = 28.5, group 1 ≠ group 2, *p* = 0.003, two-tailed), and group 2 pigs (median = 1) and their control lambs (median = 0) (Mann–Whitney U = 2, group 1 ≠ group 2, *p* = 0.01, two-tailed), were nonetheless observed. Among the samples selected for IHC testing, suckling piglets (group 1: n = 3 and group 2: n = 2), and weaners (group 1: n = 2) showed positive staining for RVFV antigens in the livers only (Figure 4). A number of livers from both groups of pigs presented with tiny scattered positive staining nonetheless. Hepatocyte glycogen storage vacuoles can, however, be normal in fed piglets [36].

### 3.3. Serology

#### 3.3.1. Pregnant Sows and Resultant Offspring

All pregnant sows (n = 5) in group 1 (M66/09 virus variant) seroconverted from 14 DPI, and remained positive until their humane euthanasia on different DPI. Anti-RVFV antibodies were detected in the offspring (n = 25; 75. 76%) of three of the sows which farrowed (Figure 5). In group 2 (M21/10 virus variant), half the sows (n = 2) and their offspring did not seroconvert, while the remaining sows (n = 2; 50%) and their piglets (n = 16; 43.2%) seroconverted from 4 DPI (Figure 5). All group 1 control lambs (n = 2) seroconverted, but group 2 control lambs did not (n = 2). The proportion of seropositives between group 1 and group 2 sows and their offspring were significantly different *(p =* 0.0015) (Appendix A).

#### 3.3.2. Suckling Piglets and Weaners

In group 1, a mock infected sucking piglet (n = 1) and weaners (n = 3), and a negative control weaner (n = 1) seroconverted. In group 2, an infected suckling piglet (n = 1) and weaners (n = 6) and a mock infected weaner (n = 1) demonstrated antibody presence. Differences in the proportion of seropositive infected suckling piglets and weaners between the two groups were not significant *(p =* 0.27). Among the animals infected with a mixture of M66/09 and M21/10 virus variants in group 3, control lambs (n = 4) and a weaner (n = 1) demonstrated antibody presence *(p =* 1.00) (Appendix A).

### 3.4. Real Time RT-PCR

#### 3.4.1. Pregnant Sows and Resultant Offspring

Rift valley fever virus RNA was detected in a few organs, oronasorectal swab pools and blood samples from both group 1 and 2 (Appendix A). No RVFV RNA was detected in the organ pool and blood samples of the infected pregnant sows in the two groups. However, in group 1, oronasorectal swab pools from two pregnant sows each tested positive on 3 and 4 DPI, and on 2 and 4 DPI, while an oronasorectal swab pool from a single pregnant sow in group 2 tested positive on 21 DPI. In group 1, organ pool samples from aborted foetuses (n = 2) and newborn piglets (n = 14) from sows (n = 4), and one blood sample, collected at 28 DPI from a newborn piglet, yielded positive results. Positive results in group 2 were obtained from organ pool samples of stillborn (n = 1) and newborn piglets (n = 19) from all infected sows (n = 4), and from blood collected on 27 DPI from a newborn piglet. Ct-values in groups 1 and 2 ranged from 18.97–39 (median: 35.8) and 23.97–38 (median: 34.15), respectively, and their median distributions were not significantly different (Mann–Whitney U = 136, group 1 ≠ group 2, *p* = 0.13). Median Ct-value distributions of group 2 sows and litters and their control lambs were significantly different (Mann–Whitney U = 0, group 1 ≠ group 2, *p* = 0.02).

#### 3.4.2. Suckling Piglets and Weaners

Five organ pools tested positive in group 1, including suckling piglets (n = 3), weaner (n = 1) and positive control lamb (n = 1). The blood of one newborn piglet and oronasorectal swab pools of a suckling piglet, weaners (n = 2) and a lamb tested positive. Group 2 recorded ten positive organ pool samples, including those from suckling piglets (n = 3), a weaner (n = 1), positive control lambs (n = 2) and uninfected lactating sows (n = 2) and ewes (n = 2). Blood of weaners (n = 2) and a control lamb (n = 1) tested positive. Oronasorectal swab pools of a suckling piglet (n = 1) and weaners (n = 4), and negative control weaner (n = 1) also tested positive in this group (Appendix A). Ct-values ranged from 15–35.22 (median: 34.47) and 26.47–33.98 (median: 32.51) in groups 1 and 2, respectively, and their distributions did not vary significantly (Mann–Whitney U = 11, group 1 ≠ group 2, *p* = 0.1). Median Ct-value distributions of group 2 porcines (median = 32.51) and their control lambs (median = 16.26) were significantly variable (Mann–Whitney U = 0, group 1 ≠ group 2, *p* = 0.04).

In group 3, organ pool samples of an infected weaner (n = 1) and an uninfected control weaner (n = 1) and a blood sample from a control lamb (n = 1) tested positive (Appendix A).

### 3.5. Virus Isolation and Conventional RT-PCR

#### 3.5.1. Pregnant Sows and Their Offspring

One to three passages were performed per sample. Atypical Vero cell morphology, which was probably CPE, was observed following inoculation with the experimental porcine samples when compared with the cell controls for approximately 50% of all the organ pools and blood tested (Figure 6). Presence of RVFV RNA in TC supernatants was only determined for organ pool samples, which consistently yielded CPE–like appearance on cell culture in subsequent blind passages. Blood samples only underwent a single passage in cell culture and the consistency of their effect on the cell monolayers was not verified. Conventional RT-PCR yielded 17/44 (38.6%) and 18/42 (42.85%) positive results for group 1 and 2 organ pool samples, respectively *(p =* 0.69).

#### 3.5.2. Suckling Piglets and Weaners

Similar to the results obtained from samples of the pregnant sows and their offspring, atypical Vero cell morphology was observed following infection with organ pool samples from groups 1 to 3, comprising ewes (n = 8), lambs (n = 8), lactating sows (n = 3), suckling piglets and weaners (n = 40) and their terminal bleeds (n = 30), in approximately 30% of the flasks. Of the samples tested by conventional RT-PCR, only one lamb from group 2 tested positive.

### 3.6. Sequencing and Phylogenetic Analysis

#### 3.6.1. RT-PCR

Among the samples tested by conventional PCR (n = 17), only 11 showed visible bands on agarose gel electrophoresis (Figure 7), of which 9 were selected for purification and sequencing since they had acceptable nucleic acid concentrations. These were group 2 control lambs (n = 2), a weaner (n = 1), a pregnant sow (n = 1) and her piglet (n = 1), group 1 piglets (n = 2) and two positive controls, i.e., TC material from Onderstepoort Biological Products (OBP) and virus, M21/10.

#### 3.6.2. Sequencing and Phylogenetic Analysis

Sequences were successfully generated for six of the nine samples, and included a group 1 piglet infected with RVFV strain M66/09 variant (n = 1), group 2 infected animals (M21/10 variant), lamb (n = 1), weaner (n = 1) and pregnant sow (n = 1), and PCR positive controls M21/10 variant (n = 1) and OBP-TC virus (n = 1). Partial and full M-segment genome sequences (n = 37) were sourced from Genbank for confirmation of identity and comparison of genotypes, bringing the total number of taxa analysed to 44 (Figure 8). End-unaligned sequences were removed, resulting in a final dataset of 353 nucleotides in length. Phylogenetic analysis revealed that sequences generated from groups 1 and 2 animals clustered within lineages C and H with the M66/09 and M21/10 variants, respectively, while the OBP tissue culture virus grouped within lineage K (Figure 8). Percent identities between sequences generated in this study and selected reference sequences from the 2009 and 2010 RVF outbreak strains ranged from 96.88% to 100% and 91.45% to 100% at nucleotide and amino acid levels, respectively.

### 3.7. Electron Microscopy

Round to icosahedral particles of 90 nm–110 nm consistent with the shape and size of RVFV were identified by negative staining of liver samples of aborted foetuses (n = 3) from a sow infected with the RVFV M66/09 variant (Figure 9).

## 4. Discussion

Successful experimental infection of pregnant sows with RVFV was achieved in this study. Sero-conversion of 5/5 (100%) and 2/4 (50%) of sows infected with two genetically diverse RVF viruses was demonstrated (Figure 5). Real time RT-PCR testing of organ pool samples of all the sows and their blood did not yield positive results, but oronasorectal swab pools confirmed the presence of RVFV RNA in two pregnant sows in group 1 and one in group 2. In addition, RVFV antigen/RNA and antibodies were demonstrated in livers, organ pools, blood and sera of the offspring of the RVFV-infected sows, attesting to successful infection of their dams, since they were not inoculated (Figure 4 and Figure 5; Appendix A).

Suckling piglets and weaners were also successfully inoculated with RVFV in this study, as shown by seroconversion in ELISA, demonstration of viral antigen in their livers and RNA in their organ pools, blood and oronasorectal swabs, using immunohistochemistry and real time RT-PCR, respectively (Figure 4 and Figure 5; Appendix A). These findings corroborated those of [16], who infected six weaners with 10^5^ pfu/mL of RVFV ZH501 strain subcutaneously, and demonstrated that whilst all seroconverted from 5 DPI onwards, RNA could not be directly detected from their sera and organs. In [16], viral genomic material was indirectly detected in sera (n = 3) collected on DPI 1 and 2 and oronasal swabs (n = 2) collected on DPI 3 and 5, following isolation in tissue culture (Mean Ct-value: 31.15). The differences in the proportion of samples positive for antibody in sera and RNA in oronasal swabs between this study and [16] were significant (*p* < 0.05) (Appendix A).

Nucleotide sequences obtained from porcine and ovine samples infected with the two distinct virus strains (M66/09 and M21/10) were shown to cluster within the lineages of the infecting virus strains (Figure 8), further confirming successful infection of the pigs with RVFV.

### 4.1. Effect on Reproduction

Reproductive failures characterised by an abortion and expulsion of normal, macerated and mummified foetuses, birth of stillborn and weak piglets and neonatal mortalities were observed. The live piglets tested RVFV antibody and antigen/RNA positive in various samples, while the aborted foetuses also tested positive for viral RNA in a few samples (Table 2; Figure 1, Figure 4 and Figure 5; Appendix A). Reproductive failures like these may result from non-infectious or infectious causes and their resultant pathogenesis, or both [37]. However, RVFV was the most likely cause because common infectious pathogens associated with stillbirth, mummies, embryonal deaths and infertility (SMEDI) were unlikely, since the pigs were sourced from a closed breeding herd with strict biosecurity and adherence to disease control regulations, conditions which are also protective against management causes. Our experimental findings support Weiss’s field observations [38] that pregnant sows aborted amidst ewe abortions during an outbreak of RVF in South Africa in the 1950s.

Vertical transmission of RVFV occurred, as evidenced by the presence of anti-RVFV antibodies in the sera of newborn piglets and through detection of viral RNA in their organ pools and blood samples. The RVFV-positive newborn piglets were from the sows that farrowed in groups 1 and 2 (Figure 5; Appendix A). The following findings provided further proof of vertical transmission of RVFV from sows to their offspring: (i) presence of viral genomic material in an organ pool of one aborted foetus; (ii) demonstration of putative RVFV particles by negative staining electron microscopy in liver samples of three foetuses in group 1; (iii) positive antigen labelling in IHC sections of livers of aborted foetuses and newborn piglets from the two groups, albeit faint due to the low concentrations of virus in the tissues as evidenced by high PCR Ct-values, (Table 4; Figure 4 and Figure 9; Appendix A). These observations also provided proof that the abortion in this study was caused by infection with RVFV. Nonetheless, vertical transmission of RVFV without demonstration of viraemia, clinical signs and seroconversion in dams and offspring does occur, as proven by [39], who demonstrated presence of viral RNA in pregnant ewes and their foetal organs, and live virus in the organs of the foetuses, similar to group 2′s PS 1 and PS 4 and their offspring (Appendix A).

Teratogenicity in piggeries, caused by hereditary factors, nutritional factors or poisons and infectious agents, is a common occurrence worldwide and incidence rates of 0.11% to 4.96% have been reported [40]. In this study, congenital defects in the aborted foetuses and newborns were observed in 9% of piglets (Table 2; Figure 1). The authors in [41] reported that mouse brain passaged and live-attenuated Smithburn vaccine strains caused abortions and teratogenic effects, including arthrogryposis, at 42 to 74 days of pregnancy in ewes [42]. It is, therefore, possible that RVFV was the cause of some of the congenital abnormalities observed in the pigs, but the phenomenon involving non-vaccine strains needs further investigation [43]. No evidence of teratogenicity was found in naturally infected ovine foetuses [34].

### 4.2. Serology

Seropositive pigs were observed, but not all infected pregnant sows, suckling piglets and weaners and newborn piglets tested positive for RVFV antibodies using the IDVET RVF Blocking ELISA kit (Figure 5; Appendix A). The negative serology results for the pigs could have been due to deposition of the virus in subcutaneous tissue instead of inside the jugular vein [44]; virus replication failure in the infected pigs [1]; dominance of cell-mediated instead of a humoral immune response [45]; absence of anti-RVFV antibodies in the colostrum and milk of the sows, and thus, no absorption of the antibodies by offspring; failure of piglets to suckle from sero-positive sows; failure of the virus to cross the placenta and infect all foetuses [46]; and development of immune tolerance by the infected foetuses. Alternatively, it could point to low levels of sensitivity of the test. Control lambs infected with M21/10 virus variant tested negative for antibodies, probably because they were euthanised on 3 DPI before mounting measurable immune responses. However, control lambs infected with the M66/09 virus variant only tested positive on 29 DPI. These combined results suggest that the cause of the majority of the negative results in both pigs and lambs is likely due to low sensitivity of the ELISA kit used in this study [47,48].

### 4.3. Polymerase Chain Reaction

Presence of viral genomic material was demonstrated in some, but not all organ pools, oronasorectal swabs and blood of infected animals and their offspring by real-time RT-PCR (Appendix A). The mandatory inactivation protocols and movement of samples from the BSL3 stable facility to the diagnostic laboratory resulted in unavoidable suboptimal sample storage conditions. In addition, the time lapse before testing [49,50], assay validation in porcine samples and reagents used [51,52], sample pooling [53,54] and presence of virus below the assay’s minimum detection range [16] could have contributed to some samples testing negative. Sample pooling most likely reduced assay sensitivity due to the dilution effect [53,54]. The pathogenesis of RVFV in the pig model has never been extensively studied and undisseminated infection in some of the inoculated pigs, gestation period at the time of infection of the pregnant sows and failure of the virus to cross the placenta of some foetuses cannot be ruled out [55].

### 4.4. Routes of Transmission

It was interesting that several oronasorectal swab pools from pigs (58.8%) in this study yielded positive results on PCR, highlighting the possibility that RVFV could be shed in the secretions and/or excretions of infected pigs (Appendix A). Based on our results, shedding is estimated to occur for at least one month, since the oronasorectal swab pool of one infected weaner in group 1 (M66/09 virus variant) was positive at 28 DPI. However, because the swabs were pooled, it was not possible to identify which excretion/secretion i.e., oral, nasal or rectal, contained the viral RNA. This, combined with the lack of virus isolation from these swabs, is a limitation of this study. Nonetheless, the results were consistent with those of other studies which reported positive RVFV PCR results from oral and nasal swabs of experimentally infected animals or isolation of virus from such samples, or both [16,35,56].

Contact transmission of RVFV via an unknown route under experimental conditions on the 7th day post exposure was first observed by [57]. Transmission of RVFV from lamb to lamb though an unclear mechanism was later described [58]. Horizontal transmission was not recorded even though virus was present in the oronasal and saliva swabs of infected animals [16,56]. In our study, viral RNA was present in the organ pools of a negative control suckling piglet (group 1), two lactating ewes and sows each (group 2) and a weaner (group 3) and from the swab of a weaner (group 2) collected five days post exposure (DPE) (Appendix A). Anti-RVFV antibodies were demonstrated in controls, lactating sow and weaner on 14 and 21 DPE in group 1, and one weaner each in groups 2 and 3 on 14 and 30 DPE, respectively (Appendix A). Contact with the infected secretions could have been the mode of transmission to the mock infected and uninfected animals. However, the combined results of our study and those conducted previously under experimental conditions, showed that RVFV can be transmitted horizontally among in-contact animals, even though the exact mechanism of transmission is not known.

### 4.5. Virus Isolation

We attempted to isolate RVFV from organ pools and terminal bleeds of the lactating and pregnant sows, aborted foetuses, newborn piglets, suckling piglets and weaners using Vero cell lines in this study. Similar to the PCR results, factors such as sample pooling, processing, storage, handling and time lapsed before testing, and the fact that RVFV infection kinetics in the pig model are unknown, could have affected the success rate of isolating virus from the majority of the samples. Consistent cell degenerative changes characterised by non-lytic cell swelling were, however, observed for a number of organ pool samples, and positive PCR results were obtained from some of the corresponding TC supernatants following two to three blind passages, suggesting some degree of virus replication (Figure 6; Appendix A).

### 4.6. Genetic Variation

Genetic analysis of the TC grown and passaged viruses, and one organ pool swab virus in this study, revealed that they clustered within 3 of the 15 different lineages identified by [59], i.e., lineage C, H and K (Figure 8). Viruses from a weaner, pregnant sow and control lamb in group 2 (infected with M21/10 virus variant), clustered within lineage H with strain M21/10. The OBP virus grouped with strains in lineage K, while a piglet born to a pregnant sow from group 1 (infected with M66/09 virus variant), clustered with strain M66/09 in lineage C.

In a previous study, full genome sequences of strains M66/09 and M21/10 at passage levels 1 to 3BHK each confirmed the M-segment clustering of these viruses within lineage C and H, respectively [21]. In this study, the viruses were further passaged in Vero cells to increase their titres before inoculating the animals, and except for one weaner sample from group 2, additional passages in Vero cells following termination of the animal experiments were completed before sequencing. They observed 0.29% and 0.86% differences in identities at nucleotide and amino acid levels, respectively, between the parental M66/09 sequence deposited in GenBank and that obtained from a newborn piglet in group 1 could be attributable to mutations arising during viral replication in the different host systems [21,60,61], and the fact that only 353 bp partial genome sequences of the Gn glycoprotein were used to compare the isolates [62]. Increased number of sequences from each infection group could have been useful in analysing clustering of viruses within the lineages, especially their relationship with parental strains, M66/09 and M21/10. Nonetheless, nucleotide percent identity differences of 0% to 5% were observed among the virus sequences used to infer phylogeny in this study, which were similar to differences observed by other workers [63], further underscoring the conserved nature of the RVFV genome.

### 4.7. Pathology

Numerous publications have reported on the macroscopic pathology of natural or experimental RVFV infections in domestic ruminants, especially sheep, where liver friability, congestion and haemorrhage and yellow/orange-brown discolouration due to diffuse necrosis or disseminated grey-white areas of necrosis were the predominant findings in neonates. Other organ systems also show signs of circulatory impairment [5,34,64,65]. Unlike in some adult ruminants, no gross post mortal changes were seen in the infected sows. However, in the aborted foetuses and newborn piglets, suckling piglets and weaners, macroscopic lesions similar to those in affected ruminants but with less severity, were observed mainly in the liver, spleen and kidneys. Blood-tinged hydrothorax, hydropericardium and ascites were seen in a few cases, especially in the aborted foetuses and sick newborn piglets, and similar observations were made in sheep foetuses and lambs [5,34]. No lesions were observed at necropsy, except for a slightly enlarged lymph node on the inoculation side in one weaner by [16].

Histopathological examinations clearly showed that RVFV infection in the ruminant neonate caused massive hepatic necrosis and haemorrhages with fatty metamorphosis and hydropic degeneration only observed among the few surviving hepatocytes [5,34,64,65]. The insult to the porcine neonate liver was mainly characterised by hydropic degeneration (cellular swelling/hydropic change/vacuolar degeneration/cellular oedema). Another contrast observed was in the kidney and spleen, where subcapsular haemorrhages featured prominently in ruminants compared to mainly congestion in the pig samples. In addition, there were striking differences observed in the spleens, where lymphocytolysis was a prominent lesion mostly in the red pulp of foetuses and lambs, and white and red pulp of adult sheep, while piglets and weaners showed white pulp expansion (Table 4) [5,34,66]. However, tubular epithelial injury without meaningful inflammation, with or without proximal tubular epithelial (PTE) cell degeneration with pyknotic nuclei and detachment of cells from the tubular basement membrane, was a common kidney lesion between the ovines and porcine kidneys [5,34,66]. The only microscopic lesions reported by [16] were mild lymphoplasmacytic perivascular cuffing and multifocal glial nodules with vacuolation in the brain neuropils of two viraemic weaners. The development of non-lipid, glycogen filled vacuoles in the hepatocytes of the infected porcine livers could underlie the apparent tolerance to infection and prevention of degenerative changes and necrosis [67,68].

## 5. Summary and Conclusions

There were clear similarities and differences in the clinico-pathological outcome of RVFV infection in the domestic pig and sheep and cattle observed in this and other studies. Similarities were that pregnant animals aborted, the virus was vertically transmitted, reproductive disorders occurred, anti-RVFV antibodies and viral RNA could be detected in offspring born from infected sows, subadult and non-pregnant animals did not display clinical signs and macroscopic lesions characteristic of RVFV infection were notable in the liver, spleen and kidneys. Inconsistencies with clinico-pathological outcomes and laboratory analysis of samples from experimentally infected animals characterised by negative results for several, but one or two analytes were common among this study and others conducted in pigs, sheep and rats. Differences were that neonatal piglets were subclinically infected, unlike their domestic ruminant counterparts, and on histopathology, liver lesions in infected pigs were mainly characterised by mild necrosis and non-lipid glycogen-filled vacuoles. This is contrary to severe pan-necrosis observed in domestic ruminant species.

It is concluded that domestic pigs can be infected with very high RVFV titres via a yet to be determined efficient route and their oronasal secretions potentially act as brief sources of virus to susceptible animals that are in close contact. The blood of infected newborn piglets and weaners can also potentially infect open human skin and wounds. It is advisable that personal protective equipment (PPE), just like with ruminants, should be used when slaughtering, assisting with farrowing-related processes and handling/performing post mortem examinations on aborted foetuses and carcasses of pigs during RVF outbreaks, in order to prevent possible pig to human transmission of the disease.

## Figures and Tables

**Figure 1 viruses-15-00545-f001:**
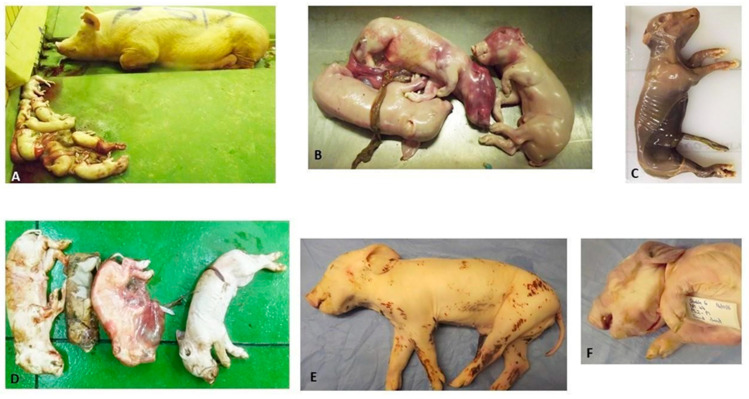
(**A**) PS5A and aborted foetuses in group 1; (**B**) Aborted foetuses presenting with congested skin over the head area and arthrogryposis; (**C**) Mummified foetus expelled by PS5A in group 1; (**D**) Dead piglets (stillborn) born from PS4C in group 2; (**E**) Group 2′s PS2-P3C with splayed left hind limb and decubitus ulcer on the medial aspect of the right hind leg; and (**F**) Group 2′s P2-P1C with brachycephalus and arthrogryposis.

**Figure 2 viruses-15-00545-f002:**
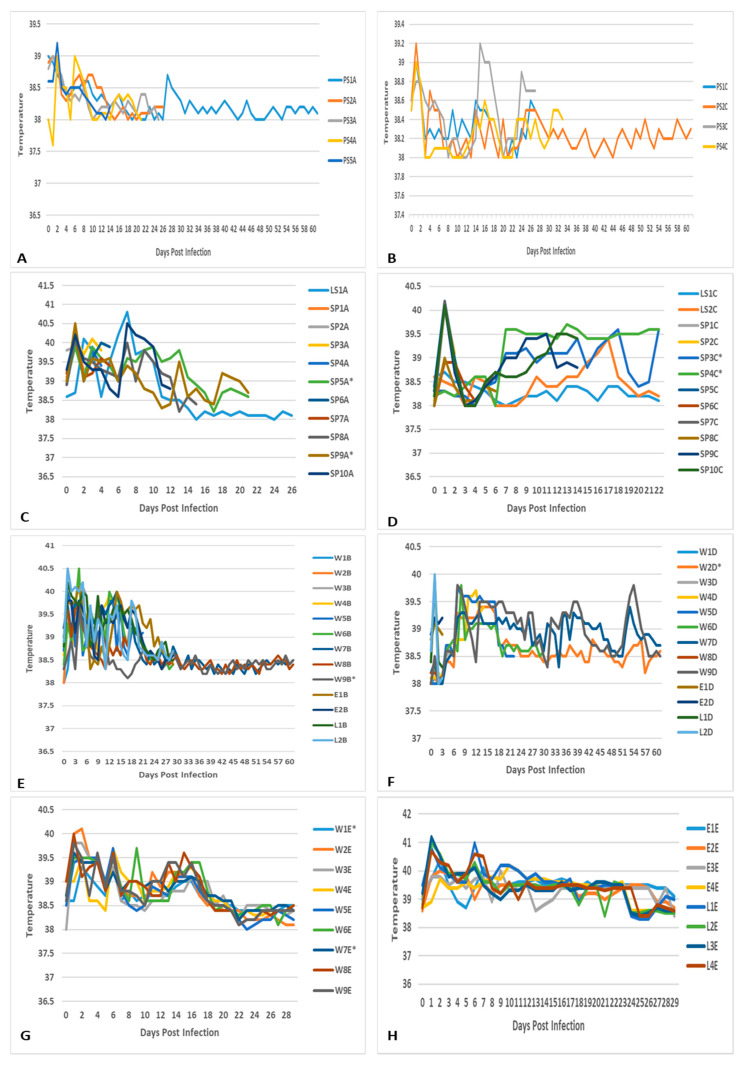
Temperatures (degrees Celsius) of: (**A**) pregnant sows (PS) infected with RVFV 1 (M66/09 variant) in group 1 from 0–61 days post infection (DPI); (**B**) pregnant sows (PS) infected with RVFV 2 (M21/10 variant) in group 2 from 0–61 days post infection (DPI); (**C**) suckling piglets (SP) infected with RVFV—M66/09 variant and lactating sow (LS) in group 1; (**D**) suckling piglets (SP) infected with RVFV—M21/10 variant and lactating sows in group 2; (**E**) weaners (W) and control lambs (L) infected with RVFV—M66/09 variant, and uninfected ewes (E) in group 1; (**F**) weaners (W) and control lambs (L) infected with RVFV—M21/10 variant, and uninfected ewes (E) in group 2; (**G**) uninfected control weaners (W; W1 and W7), those infected with RVFV M66/09 variant (W2 and W5) and M21/10 variant (W4 and W6) only, and those infected with a mixture of the two viruses (W3, W8 and W9) in group 3; (**H**) control lambs (L) infected with RVFV mixture (M66/09 and M21/10 variants) and uninfected ewes (E) in group 3. *** Denotes control animal.

**Figure 3 viruses-15-00545-f003:**
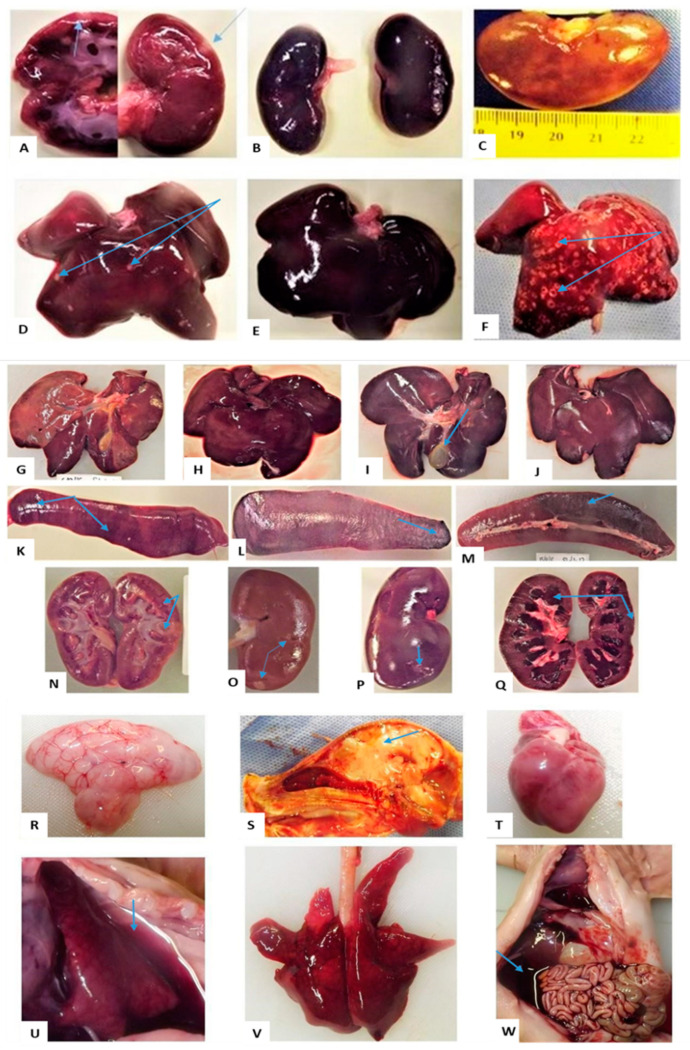
Suffixes A and B (group 1), C and D (group 2) and E (group 3), at the end of each animal ID, denote their stable numbers, and lesions are pointed to with blue arrows: (**A**) Infarct in cortex and medulla of PS5A-AF1 kidney; (**B**) Congestion and haemorrhage of PS5A-AF3′s kidneys; (**C**) Soft, enlarged kidney of PS2C-P9 with diffuse pale-yellowish areas; (**D**) Infarcts on the liver of PS5A-AF2; (**E**) Congestion and haemorrhage of PS5A-AF3′s liver; (**F**) Multifocal necrotic foci on PS5A-AF10′s liver giving it nutmeg appearance; (**G**) Congested liver of SP5C on DPI 4; (**H**) Congested liver of SP6A at DPI 6; (**I)** Congested liver of W4D on DPI 15, with oedematous gall bladder; (**J**) Congested liver of W3B on DPI 6; (**K**) Enlarged spleen of SP5C with haemorrhages; (**L**) Enlarged spleen of W4D on DPI 15 with haemorrhages; (**M**) Spleen of W3B on DPI 6 with haemorrhages; (**N**) Kidney of SP7A on DPI 6 with infarcts; (**O**) Kidney of SP8C on DPI 15 with infarcts; (**P**) Kidney of W4D in on DPI15, with infarcts and a rough cortical surface remaining following peeling of the capsule; (**Q**) Severe congestion and tubular degeneration of a kidney of W3B on DPI6; (**R**) Soft and jelly-like brain of PS5A-AF1; (**S**) Brain of PS5A-AF1 with unclear delineation between white and grey matter; (**T**) Soft and pale heart of PS5A-AF2; (**U**) PS5A-AF3′s collapsed congested lung and blood-tinged hydrothorax; (**V**) Oedema, congestion and haemorrhage of PS5A-AF1′s lung; (**W**) Blood-tinged ascites in PS5A-AF2 and congested GIT.

**Figure 4 viruses-15-00545-f004:**
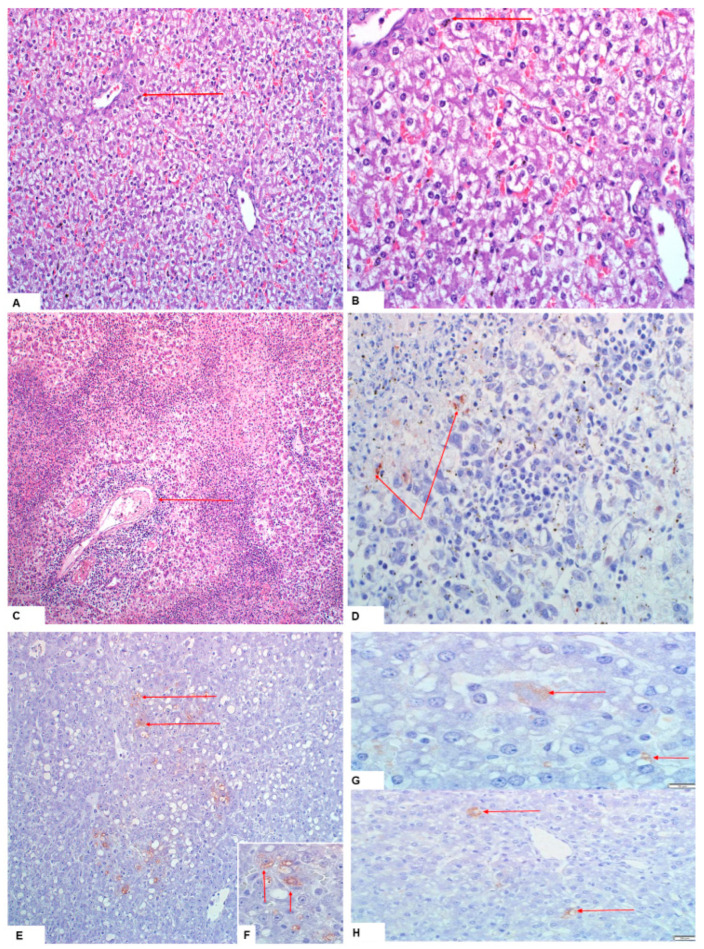
Lesions and RVFV antigen are indicated with red arrows: (**A**,**B**) hepatocyte swelling with clarification of the cytoplasm of a neonate piglet born from a pregnant sow infected with M21/10 virus variant in stable C (H&E staining, 20× and 40× magnification); (**C**,**D**) necrotic hepatocytes and infiltration of inflammatory cells around the hepatic portal vein of a PS5 foetus, and faint RVFV antigen foci on IHC staining of the same foetus from the M66/09 virus variant infected sow; (**E**,**F**) positive antigen staining in liver of a neonate from a PS (the M66/09 virus) in group 1 (20× magnification and close-up view, respectively); and (**G**,**H**) Liver of SP2A (the M21/10 virus) showing RVFV 1 antigens in the hepatocytes (100× and 40× magnification, respectively).

**Figure 5 viruses-15-00545-f005:**
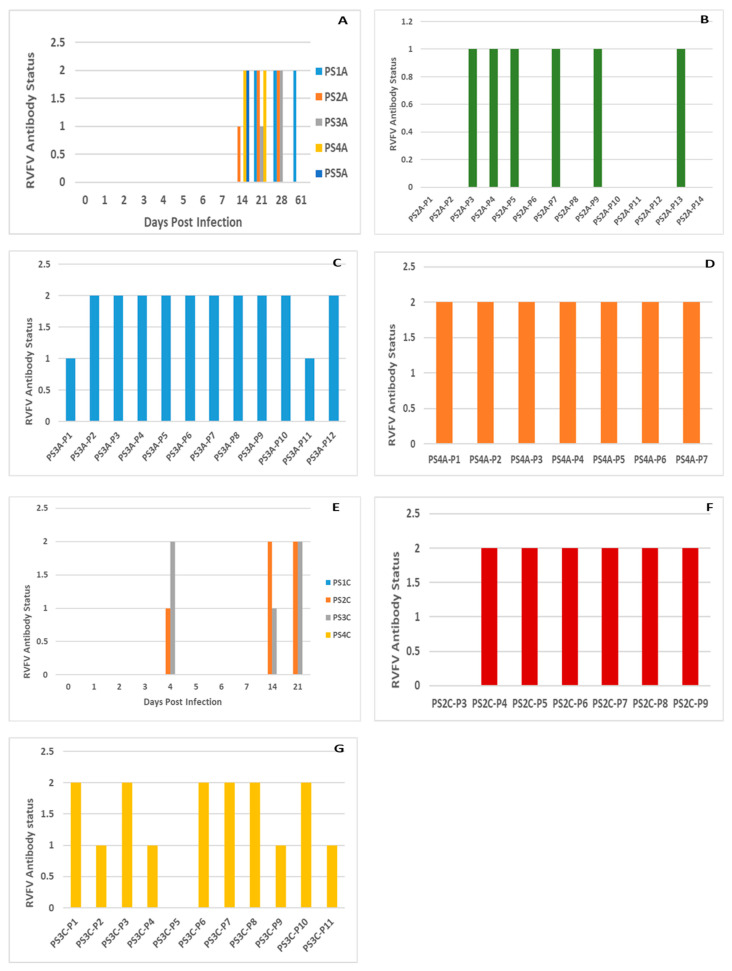
A: IDVet Blocking ELISA detected antibody statuses of: (**A**) Pregnant sows (PS) infected with RVFV1 in stable A (S/N% mean: 16.40); (**B**–**D**) Piglets born from PS2A (S/N% mean: 53.73), PS3A (S/N% mean: 32.17) and PS4A (S/N% mean: 27.57) in stable A on 23, 28 and 32 DPI, respectively; (**E**) PS infected with RVFV 2 in stable C (S/N% mean: 57); (**F**) Piglets born from PS2C (S/N% mean: 33.85) and (**G**) PS3C (S/N% mean: 42) in stable C on 44 and 22 DPI, respectively. Antibody statuses of 0, 1 and 2 denote negative, suspect and positive, respectively.

**Figure 6 viruses-15-00545-f006:**
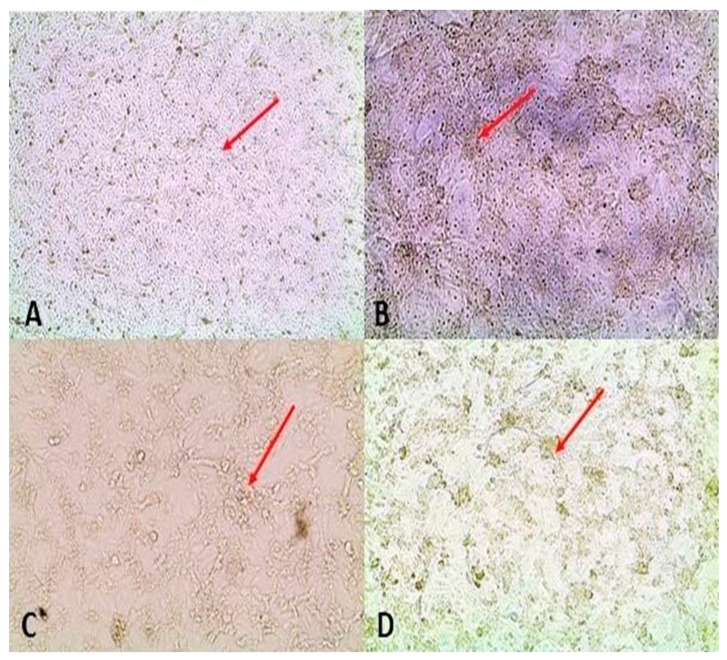
Red arrows point at normal cell monolayers and cytopathic effect: (**A**) Three-day-old control Vero cells maintained with the same medium as the one used to isolate virus from pooled organ homogenate supernatants; (**B**) Day 3 of neonate (group 1) organ pool passage 1 on Vero cells; (**C**) Day 3 of neonate (group 1) organ pool passage 1 on Vero cells; (**D**) Day 3 of neonate (group 2) organ pool passage 1 on Vero cells.

**Figure 7 viruses-15-00545-f007:**
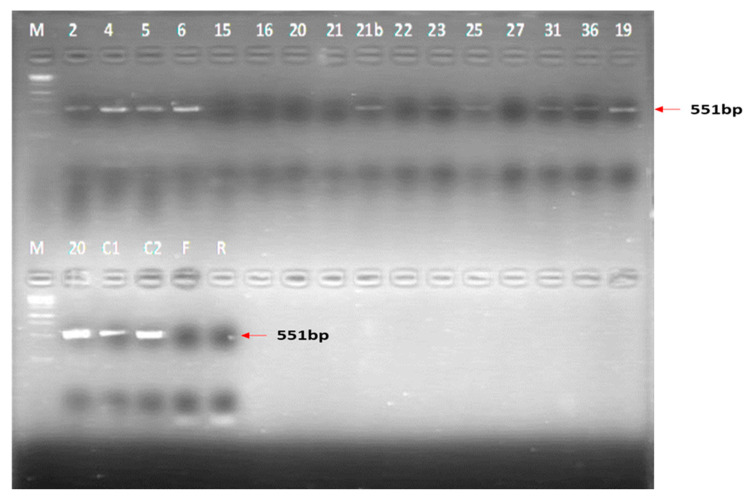
Results of the conventional RT-PCR run which informed the choice of samples to analyse for purity and sequencing. M = 1 kb marker; 2 = PS2C Piglets (group 2); 4 = L1D (group 2); 5 = PS4C Piglets (group 2); 6 = PS2C (group 2); 15 = PS5A Foetuses (group 1); 16 = PS1A (group 1); 20 = PS2A (group 1); 21 = PS3A (group 1); 21b = W4D (group 2); 22 = PS1C (group 2); 23 = PS2A Piglets (group 1); 25 = PS3C (group 2); 27 = W1B (group 1); 31 = PS1C Piglets (group 2); 36 = PS3A Piglets (group 1); 19 = L1D (group 2); C1 = M21/10 virus Control; C2 = OBP TC positive control; F = Forward primer; R = Reverse primer.

**Figure 8 viruses-15-00545-f008:**
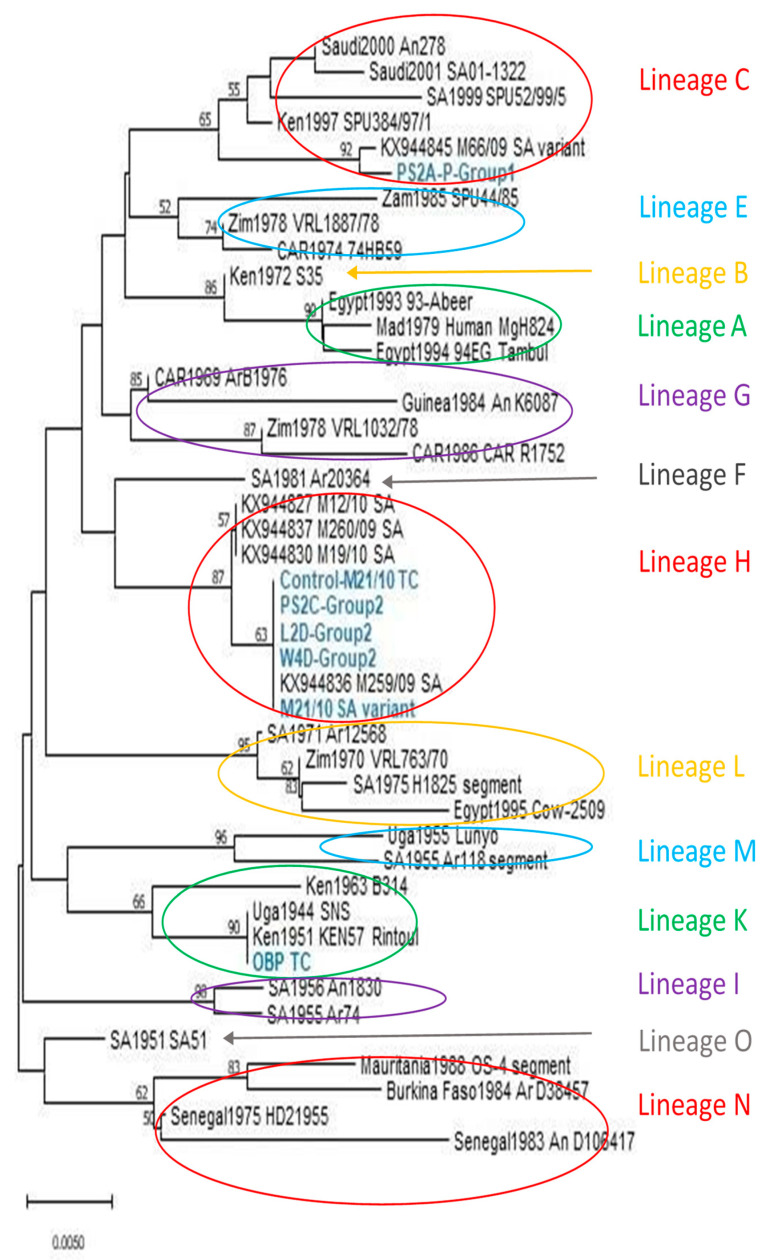
Neighbour-joining tree showing clustering based on partial Gn glycoprotein sequences generated for the C, H and K lineage viruses characterised in this study (taxa indicated in blue and bold). Nucleotide distances were computed using the Maximum Composite Likelihood method and nodal support was tested through 10,000 non-parametric bootstrap replications. Bootstrap values above 50 are shown above the branches.

**Figure 9 viruses-15-00545-f009:**
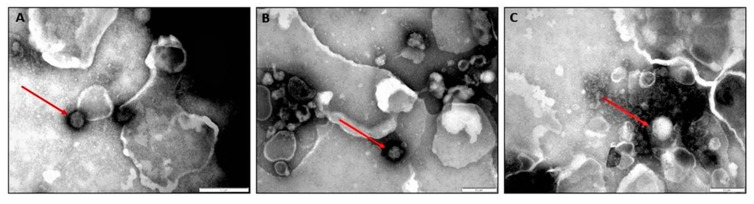
(**A**–**C**) Negative staining electron microscopic images of livers of three foetuses showing putative RVF virions of approximately 100–120 nm (denoted with red arrows). The image scale is 200 nm.

**Table 1 viruses-15-00545-t001:** Clinical score descriptions of infected pigs and control lambs.

Clinical Score	Description
0	No clinical signs
1	Pyrexia: ≥40 °C
2	Small size and weakness (porcine): decreased appetite, listlessness and disinclination to move, Weakness (ovine): decreased appetite, listlessness and disinclination to move
3	Alive with splay legs, arthrogryposes, umbilical hernia and other abnormalities (porcine neonates) Disinclination to move, anorexia and weakness (suckling ovines and porcines and weaners)
4	Neonatal mortality; stillborn; mummies, fresh and macerated foetuses with or without abnormalities
5	Abortion (sows)

**Table 2 viruses-15-00545-t002:** Comparison of clinical scores of pregnant sows and their offspring, suckling piglets and weaners, and weaners in groups 1 (M66/09 variant), 2 (M21/10 variant) and 3 (mixture of M66/09 and M21/10 variants), and their corresponding control lambs. Negative control animals were excluded from scoring.

M66/09 Virus Variant Inoculated Animals	No. of Animals (n)	Ave. Score	Observations
PS1A	0	0	N/A
PS2A and piglets	17	0.375	Weakness and lameness (n =1); umbilical hernia (n = 1); normal (n = 15)
PS3A and piglets	15	0.67	Small and weak (n = 1); dead (n = 2); normal (n = 12)
PS4A and piglets	8	0	Sow and all piglets healthy
PS5A and foetuses	21	4.05	Abortion (n = 1); dead foetuses (n = 14, of which two had arthrogryposis); mummified foetuses (n = 2); macerated foetuses (n = 4)
Average score	1.273		
SP1A, SP2A, SP3A, SP4A, SP6A, SP7A, SP8A and SP10A	8	0.75	Pyrexia (n = 6); normal (n = 2)
Average score	0.75		
W1B, W2B, W3B, W4B, W5B, W6B, W7B, and W8B	8	0.5	Pyrexia (n = 4); normal (n = 4)
Average score	0.5		
L1B	N/A	1	Pyrexia
L2B	N/A	1	Pyrexia
Average score	1		
M21/10 virus variant inoculated animals	No. of animals (n)	Ave. score	Observations
PS1C and piglets	18	1222	Weak (n = 1); dead (n = 5); normal (n = 12)
PS2C and piglets	10	1.7	Dead (n = 2; all with arthrogryposis and 1 with brachycephalus); alive with splay legs (n = 3); normal (n = 5)
PS3C and piglets	12	0.167	Weak (n = 1); normal (n = 11)
PS4C, stillborns and piglets	9	1.78	Stillborn (n = 4); normal (n = 5)
Average score	1.22		
SP1C, SP2C, SP5C, SP6C, SP7C, SP8C, SP9C, and SP10C	8	0.75	Pyrexia (n = 6); normal (n = 2)
Average score	0.75		
W1D, W3D, W4D, W5D, W6D, W7D, W8D, and W9D	8	0	All normal
Average score	0		
L1D	N/A	3	Disinclination to move, anorexia, weak
L1D	N/A	3	Disinclination to move, anorexia, weak
Average score	3		
M66/09 and M21/10 virus variant mix inoculated animals	No. of animals (n)	Ave. score	Observations
W3E	N/A	1	Pyrexia
W8E	N/A	0	None
W9E	N/A	0	None
M66/09 and M21/10 virus variant mix inoculated animals	No. of animals (n)	Ave. score	Observations
Average score	0.33		
L1E	N/A	1	Pyrexia
L2E	N/A	1	Pyrexia
L3E	N/A	1	Pyrexia
L4E	N/A	1	Pyrexia
Average score	1		

N/A: Not applicable.

**Table 3 viruses-15-00545-t003:** Histopathological score description. Adopted and modified from [35].

Histopathology Score	Description
0	No lesions attributable to Rift valley fever virus
1	Multifocal, mid-zonal to central foci of lymphohistiocytic (lymphocytes and macrophages) inflammation with or without presence of few plasma cells and a single case of hepatocyte necrosis (ovines). Hepatocyte swelling with or without presence of lymphohistiocytic (lymphocytes and macrophages) inflammation, few plasma cells and a single case of hepatocyte necrosis (ovines and porcines).
2	Multifocal, 1–2 mm areas of mid-zonal to central lymphohistiocytic inflammation with central necrosis shifting inflammation to predominantly neutrophils. Involves less than 5% of the examined parenchyma.
3	Multifocal, 1–2 mm areas of mid-zonal to central lymphohistiocytic inflammation, with central necrosis shifting inflammation to predominantly neutrophils. Involves approximately 15% of the examined parenchyma, and scattered hepatocyte apoptosis is present.
4	Greater than 15% of the parenchyma is necrotic and severe multifocal haemorrhage is also present.

**Table 4 viruses-15-00545-t004:** Comparison of histopathological scores of livers from animals infected with virus or born from sows infected with virus in group 1 (M66/09 variant), group 2 (M21/10 variant) and group 3 (mixture of M66/09 and M21/10 variants). IHC scores means at least one liver in the group was positive. Lesions were either mild, moderate or severe.

Animal I.D	No. of Animals (n)	DPI	Average H-Score	Observations	IHC	H—Other Organs
Group 1: Virus variant M66/09						
PS	1	61	1	Hepatocyte swelling (glycogen) and steatosis	NT	-
PS-AF	11	14	1	Hepatocyte vacuolation, swelling (glycogen) and steatosis	+	+k: Congestion (n =4)
P	34	23–32	0.85	Hepatocyte vacuolation, swelling (glycogen) and steatosis; bile stasis; congestion; and widespread single cell necrosis	+	+k: Congestion (n = 3) +s: Congestion (n = 2)
SP	8	2–15	1	Hepatocyte swelling (glycogen) and steatosis	+	+s: Congestion (n = 1) White pulp expansion (n = 1)
W	7	2–61	1	Hepatocyte swelling (glycogen) with or without leucostasis (polymorphonuclear and mononuclear)	+	+s: White pulp expansion and congestion (n = 8) +k: Acute tubular injury; patchy PTE cell degeneration with pyknotic nuclei and detachment of cells from the tubular basement membrane (n = 3). Marked infiltrate of lymphoplasmacytic cells in the medullary interstitium (n = 1)
L	2	29	0	No lesions attributable to RVFV infection in ovines	-	-
Animal I.D	No. of animals (n)	DPI	Average H-Score	Observations	IHC	H—Other organs
Group 2: Virus variant M21/10						
PS	1	27	1	Hepatocyte swelling (glycogen) and steatosis	NT	-
SB	2	32	1	Hepatocyte vacuolation and swelling (glycogen).	NT	+k: Congestion
P	41	22–44	0.95	Hepatocyte vacuolation, swelling (glycogen) and steatosis; bile stasis; congestion; and wide spread single cell necrosis	-	+s: Congestions and haemosiderosis (n = 1) Congestion (n = 4) +k: Congestion and proximal tubular cell swelling (n = 2) Congestion (n = 2)
SP	6	2–22	1	Hepatocyte swelling (glycogen) and steatosis, with or without leucostasis (mononuclear); increased number of Kupffer cells in the sinusoids	+	-
W	7	2–62	1	Hepatocyte swelling (glycogen) and steatosis	+	+s: White pulp expansion with or without congestion (n = 9) +k: Patchy PTE cell degeneration with pyknotic nuclei and detachment of cells from the tubular basement membrane (n = 4)
Animal I.D	No. of animals (n)	DPI	Average H-Score	Observations	IHC	H—Other organs
Group 2: Virus variant M21/10						
L	2	3	2.50	Random foci of necrosis with marked infiltrate of Kupffer cells and very few neutrophils; severe necrosis involving more than 75% of the specimen with an inflammatory infiltrate of Kupffer cells and very few neutrophils, and typical RVF primary foci and nuclear inclusions	+	+s: Marked infiltrate of neutrophils in the red pulp (n = 1) +k: Subtle injury and loss of nuclei and pyknosis in the glomeruli and mild tubular injury with a few pyknotic nuclei and scattered detachment of cells from the tubular basement membrane of a few proximal tubules. (n = 1)
Group 3: Virus mix (M66/09 and M21/10)						
W	3	29	1	Hepatocyte swelling (glycogen)	-	+k: Tubular injury with or without proximal tubular epithelial (PTE) cell degeneration with pyknotic nuclei and detachment of cells from the tubular basement membrane, and infiltration of lymphoplasmacytic cells in the medullary interstitium (n = 3) +s: marked white pulp expansion, with mild to severe congestion (n = 3)
Animal I.D	No. of animals (n)	DPI	Average H-Score	Observations	IHC	H—Other organs
Group 3: Virus mix (M66/09 and M21/10)						
L	4	29	1.25	Hydropic degeneration of hepatocytes with randomly scattered lymphocytes and neutrophils	NT	+s: Congestion (n = 2) +k: Mild interstitial nephritis (n = 2).

ID: Identity; H: Histopathology; IHC: Immunohistochemistry; DPI: Days post infection; PS: Pregnant sow; AF: Aborted foetus; P: Piglet (newborn); SP: Suckling piglet; W: Weaner; SB: Still born; L: Lamb; NT: Not tested; s: Spleen; k: Kidney; +: Positive; -: Negative.

## Data Availability

Publicly available sequence datasets were analysed in this study. The datasets can be found online [69,70]. The sequences generated in this study are available on request from the corresponding author. They will be publicly available following article publication.

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
