# Peer review of "Experimental Infection of Domestic Pigs (*Sus scrofa*) with Rift Valley Fever Virus"

_viruses, 2023, doi:10.3390/v15020545_

Round 1

Reviewer 1 Report

The manuscript entitled “Experimental Infection of Domestic Pigs (Sus Scrofa) with Rift

Valley Fever Virus” (Baratang Lubisi et al) studied the susceptibility of the domestic pigs with RVFV. For this propose, the authors carried out an experimental infection of pregnant sows at different gestation periods, neonatal piglets (1-3 Days), and weaners. Clinical signs were monitored during 60 days and sera, organ pools and blood were collected to be analyzed by different techniques. Authors conclude that pigs can be experimentally infected with RVFV. Specific comments are below:

Line 85: Authors should explain more in detail why they use two different isolates of RVFV and a mix of them to infect the animals.

Animals were inoculated using intravenous route to force the infection. Because of that, some animals showed clinical signs and died of the disease. Despite of that, authors only can cause subclinical infection in some animals. It would have been interesting to infect animals using other inoculation routes. Authors should explain more in detail the reasons because they do that.

Line 473:

Authors tried to isolate virus from organ and blood samples in Vero cells. Authors didn’t observe a clear cytopathic effect in the monolayer after several passages. Instead of that, authors observe a morphological change in Vero cells concluding that they were infected. To demonstrate that, why authors didn’t carry out an immunofluorescence or other technique to detect viral antigen. Please, explain in detail.

Author Response

Dear Reviewer 1

Thank you for taking your valuable time to reviewer our manuscript. Your comments, questions and suggestions were really useful in improving the article.

Please see the attachment for our responses.

Kind regards,

Reviewer 2 Report

Overall, I believe this study and the associated manuscript is more than adequate in describing the clinical aspects of RVFV in pigs. The study clarifies the possible significance of pigs in RFV epidemiology and outbreaks.  Clearly there is still a need for significant real world outbreak investigations to test this in the non-laboratory setting but these studies are always a good first step.

I have a few minor comments below.

Line 43: The broad statement that vaccination is the only sure solution is debatable. There are cost effective vector control programs for landscape level control of mosquitoes. These are largely “impractical” often due to government regulations but during a large outbreak or in the predicted impending outbreak larvicidal treatments of mosquito breeding habitats can greatly reduce populations. Vaccination is expensive compared to many of the long lasting larvicides that can be applied to flood waters. These methods have been used repeatedly throughout the world to control mosquitoes. The following citation is one example of a spray operations that covered an area larger than some countries (Qualls WA, Breidenbaugh MS. 2020. Texas mosquito control response following Hurricane Harvey. J Am Mosq Cont Assoc 36(2S):61–67). The cost per acre of pesticides is far less than that of per head vaccination.

Line 52: I would suggest deleting the word “from other susceptible vertebrates”.  I realize your point is to focus on why you studied the pigs but the host of the virus that supports transmission could be an invertebrate if you include vectors.  This is just a suggestion.

There are some inconsistencies in how citations are written in the document. For example Line 29 and Line 73.  In the first case you just put a number in the sentence and int eh second you use the Author and then the number. Personally, the example on line 73 is far more readable but regardless check for consistency.

I am slightly uncomfortable with the clinical scoring criteria.  Pyrexia for example has a wide range from barely detectable to nearly fatal and weakness is somewhat subjective.  I would like if you could give a bit more detail in the table since tables often stand alone.

I would check throughout the manuscript for “Rift Valley fever virus” or ““Rift Valley fever” I am fairly sure that “Rift valley fever” or “Rift Valley Fever” are incorrect grammatically since the fever and virus are not location names.

Line 745-746 implies that pigs are ruminant i.e. “other domestic ruminants”.  I think you want to delete the word “other”.

Author Response

Dear Reviewer 2

Thank you for taking your valuable time to review our manuscript. Your comments, questions and suggestions were really useful in improving the article.

Please see the attachment for our responses.

Kind regards,
